# In-situ probing of the Fischer-Tropsch reaction on Co single crystal surfaces up to 1 bar

Patrick Lömker [1,2,3] ✉, David Degerman[1], Christopher M. Goodwin [1,4], Mikhail Shipilin[1], Peter Amann[1,10], Gabriel L. S. Rodrigues [1], Fernando Garcia-Martinez[3], Raffael Rameshan[5], Jörgen Gladh [1,6], Hsin-Yi Wang[1], Markus Soldemo [1], Alexander Holm [1,6,7], Steffen Tober [8,9], Jan-Christian Schober[8], Leon Jacobse[8,11], Vedran Vonk [8], Robert Gleißner[8], Heshmat Noei[8], Zoltan Hegedues [3], Andreas Stierle [8,9], Christoph Schlueter [3] & Anders Nilsson [1,2] ✉

The surface chemistry of the Fischer-Tropsch catalytic reaction over Co has still several unknowns. Here, we report an in-situ X-ray photoelectron spectroscopy study of Co(0001) and Co(10$\bar{1}$4), and in-situ high energy surface X-ray diffraction of Co(0001), during the Fischer-Tropsch reaction at 0.15 bar - 1 bar and 406 K - 548 K in a $H_2$/CO gas mixture. We find that these Co surfaces remain metallic under all conditions and that the coverage of chemisorbed species ranges from 0.4–1.7 monolayers depending on pressure and temperature. The adsorbates include CO on-top, C/-$C_xH_y$ and various longer hydrocarbon molecules, indicating a rate-limiting direct CO dissociation pathway and that only hydrocarbon species participate in the chain growth. The accumulation of hydrocarbon species points to the termination step being rate-limiting also. Furthermore, we demonstrate that the intermediate surface species are highly dynamic, appearing and disappearing with time delays after rapid changes in the reactants' composition.

The Fischer-Tropsch (FT) reaction is an important industrial process, as it produces higher hydrocarbons from synthesis gas (syngas, ≈1:2 CO:$H_2$ gas mixture) over Co, is an important industrial process[1]. The FT reaction was used during earlier time as a way to avoid oil embargos for some countries during World War II and the Apartheid regime in South Africa. In the current era it can become an important avenue for a sustainable chemical industry if CO is generated from $CO_2$ via the reverse water gas shift reaction where the $CO_2$ has been captured either directly from the atmosphere or at an intense carbon source. The hydrogen can be produced, not from the current steam reforming process of methane, but instead through electrolysis of water where the electricity is coming from a renewable source such as wind and

[1]Department of Physics, Stockholm University, 10691 Stockholm, Sweden. [2]Wallenberg Initiative Materials Science for Sustainability, Department of Physics, Stockholm University, 114 28 Stockholm, Sweden. [3]Photon Science, Deutsches Elektronen-Synchrotron DESY, 22607 Hamburg, Germany. [4]ALBA Synchrotron Light Facility, Carrer de la Llum 2-26, 08290 Cerdanyola del Vallés, Barcelona, Spain. [5]Lehrstuhl für Physikalische Chemie, Montanuniversität Leoben, 8700 Leoben, Austria. [6]PULSE Institute, SLAC National Accelerator Laboratory, Menlo Park, 94305 California, CA, USA. [7]Laboratory of Organic Electronics, Department of Science and Technology (ITN), Linköping University, SE-60174 Norrköping, Sweden. [8]Centre for X-Ray and Nanoscience CXNS, Deutsches Elektronen-Synchrotron DESY, 22607, Hamburg, Germany. [9]Physics Department, University of Hamburg, 20148 Hamburg, Germany. [10]Present address: Eduard-Zintl-Institute of Inorganic and Physical Chemistry, Technical University of Darmstadt, Peter-Grünberg-Str. 8, 64287 Darmstadt, Germany. [11]Present address: Department of Interface Science, Fritz Haber Institute of the Max Planck Society, Faradayweg 4-6, 141 95 Berlin, Germany. ✉e-mail: patrick.loemker@fysik.su.se; andersn@fysik.su.se

solar. Presently, the Fischer-Tropsch reaction utilizes Fe, Ru or Co-based catalysts that yield different hydrocarbon distributions (i.e., with regard to the abundance of shorter or longer C-chains in the product stream). Depending on the material the reactions follow these main pathways:

$$nCO + (2n+1)H_2 \rightarrow C_nH_{2n+2} + nH_2O$$

$$nCO + (2n)H_2 \rightarrow C_nH_{2n} + nH_2O$$

Further, the created water can react with the CO by the water gas shift reaction:

$$nCO + nH_2O \leftrightarrow nH_2 + nCO_2$$

The latter reaction creates more $H_2$ at the expense of CO, but on Co catalysts it is not of significant importance thus the gas mixture of 1:2 $CO:H_2$ is commonly employed[1]. The Co-based FT reaction typically generates long-chain hydrocarbons and waxes and operates at a temperature of 470-510 K and pressures of a few tens of bars[1].

The chemical state of the Co catalyst has previously been investigated through post-reaction analysis of single crystal surfaces to be in a metallic state[2–4]. However, bulk-sensitive measurements under high temperatures and pressures during *operando* of the FT reaction of Co nanoparticles have shown the existence of small amounts of oxides[5–10]. Furthermore, it has also been proposed that a partially oxidized Co catalyst can be responsible for a high activity[11]. Although no *operando* measurements during the FT reaction has detected any major presence of a Co carbide bulk phase it has been demonstrated that $CoC_2$ nano prisms shows a high selectivity to olefin formation[12]. Recent in-situ surface sensitive measurements of the FT reaction on Fe show a growing carbide phase starting immediately after the reaction is initiated[13] and on Ni at low temperatures dissolution of carbon into the bulk as a dilute carbide phase has been observed[14]. An open key question is if the state of the Co catalyst in the surface region remains fully in a metallic state or if surface oxide and near surface carbide can be present during the reaction conditions. Addressing this question necessitates detection using surface sensitive techniques performed while the reaction is turning over.

The reaction mechanism of the FT reaction consists of a sequence of elementary reaction steps[15]. The first step after CO adsorption is the dissociation of CO generating carbon monomeric species. Afterwards such C can both attach to other carbon atoms as well as adsorbed hydrogen and thus grow the hydrocarbon chain. The final step is the termination of the growth through the attachment to hydrogen atoms that results in enough weakening of the bond between the carbons and the surface, ultimately leading to desorption. The CO activation has resulted in two major hypotheses based on theoretical calculations: there is either a direct dissociation, often denoted carbide mechanism[16,17], or hydrogen-assisted dissociation via the generation of a $CH_xO$ species[18,19]. It has been proposed that the hydrogenation of adsorbed C[20] and the termination step are partly rate limiting[21] as well as hydrogenation of atomic O and OH[22,23]. Here it would be essential to probe the adsorbates on the surface, to determine intermediates that accumulate as the reaction proceeds, as a pointer towards specific rate-limiting steps.

All chemical sensitive studies over the FT reaction of Co under *operando* conditions have been conducted with methods mostly probing the bulk, such as X-ray absorption spectroscopy (XAS) and X-ray powder diffraction (XRD)[5–10]. There have been efforts to detect adsorbed species with Infrared Spectroscopy but their observation exclusively showed adsorbed CO[24] or hydrocarbons that were likely not on the Co surface[25]. Scanning tunneling microscopy (STM) have probed Co single crystal surfaces under FT at atmospheric conditions where the morphology of steps and terraces could be followed but without direct sensitivity towards the reaction intermediates and adsorbates[3,4]. However, the observed smoothness of the surface in the STM studies indirectly infers that no large rearrangement of substrate atoms has occurred related to oxide or carbide formation. In one STM study conducted at 4 bar and 492 K on the Co(0001) surface stripes were observed during the FT reaction interpreted as the appearance of long chain hydrocarbon molecules[26]. A number of surface science studies of model molecules under vacuum have been conducted on Co single crystal surfaces[22,27–29] but it is unclear if the model molecules are relevant for reactions occurring at many orders of magnitude higher pressures and temperatures.

X-ray photoelectron spectroscopy (XPS) is a unique surface sensitive method to investigate the chemical state of catalytic surfaces and adsorbed intermediates through core-level shifts. The high inelastic scattering cross-section of photoelectrons in the gas phase makes vacuum conditions necessary. Post analysis with XPS has been conducted of Co single crystal surfaces that have been in a reactor with atmospheric pressure[3,4] or 4 bar[2], at temperatures where the reaction is turning-over, followed by evacuating the reactor to vacuum and then transferring the sample to the spectrometer chamber, where the measurement was conducted. Although adsorbed CO, adsorbed carbidic carbon and hydrocarbon species were observed it is unclear if species may decompose or desorb when the system is evacuated and the temperature reduced. Near-ambient XPS (NAPXPS) studies of Co foil have been restricted to 0.1 mbar[23] − far from the conditions of atmospheric pressure where the FT reaction occurs. These studies have detected significant oxidation of the Co foil at low temperatures, while atmospheric pressure single crystal studies showed the production of methane and other short-chain alkanes and alkenes[3,4,30].

Here, we used an ambient-pressure XPS (APXPS) instrument called POLARIS operating at pressures up to 1 bar for $CO/H_2$ mixtures and as high temperatures as 506 K. The POLARIS instrument is based on the virtual pressure cell, where we create a ~30 micron thick local high-pressure cushion and utilize grazing incidence of the incoming hard X-rays to provide surface sensitivity, despite high kinetic energy of the photoelectrons[31]. The combined effect of X-ray penetration depth and electron inelastic mean free path yields an effective inelastic mean free path comparable with laboratory XPS systems of about 1.4 nm at the C $1s$ core-level and 1.3 nm at the Co $2p$ core-level[13]. The virtual pressure cell is established by introducing a high-velocity gas jet onto the catalyst and building up a dynamic pressure, such that the gas in contact with the catalyst typically interacts roughly for times on the millisecond scale. This in turn brings the FT reaction over Co into a early steady state, far away from the chemical equilibrium, with low concentrations of products in the effluent gas stream. We have used flat Co(0001) and stepped Co($10\bar{1}4$) single crystal substrates that have been shown previously to turn-over the FT reaction towards mainly methane but also minor fractions of $C_{2+}$ hydrocarbon species at close to 1 bar and 500 K[4]. Since the FT reaction is known to be structure sensitive[32–35] (i.e. a Co stepped crystal, Co($10\bar{1}15$) gives much higher turn-over than the terrace surface[3]) we have thus directly compared the Co(0001) with the Co($10\bar{1}4$) surface to elucidate the influence of steps on the reaction. In particular, the size-dependent effects that show high activity for certain size nanoparticles have in the 2010s been shown to be linked to the relative abundance of B5 sites that appear at the intersection of threefold and fourfold coordinated sites[34,36–39]. These sites can be found on the Co($10\bar{1}4$) stepped surface and thus further the explanation why stepped surfaces are observed with higher activities (See Supplementary Information S6 and Supplementary fig. 9). As FT reactions have been demonstrated at the same conditions as in the current study, we will denote the experiments as in-situ. Furthermore, we show the facile appearance and disappearance of $C_xH_y$ adsorbates as seen in the last subheading in Results and

Discussion as indicators of a state where the reaction should occur. Our complementary in-situ surface X-ray diffraction experiments yield atomic surface structure information under reaction conditions.

## Results

### Chemical and Structural State of Co Single Crystals and Adsorbates at 1 bar

First, we address the chemical state of Co: whether it is metallic, oxidic or carbidic in the near-surface region. This information would not necessary be observable in bulk-sensitive measurements, as this active phase exists only close to the surface (i.e. the first few monolayers). Figure 1a-c shows the in-situ Co $2p_{3/2}$, C $1s$, and O $1s$ signal of the

Co(0001) catalyst at a pressure of 1 bar with a reaction mixture of 1:2 $CO:H_2$ and a temperature of 406 K and 506 K (the higher corresponding to the typical FT high yield conditions) measured in the POLARIS instrument at a photon energy of 4600 eV (for samples, gases and experimental setup turn to Methods). In this work we apply the following procedure to justify the application of a peak in the spectra: The peak needs to be clearly visible in at least one instance of our measurements to be appended to the peak model. These individual observations we compile into a global fit model which we use for all spectra. We apply this global fit model to each series of spectra such that the defining peak shapes are common to all spectra as described below.

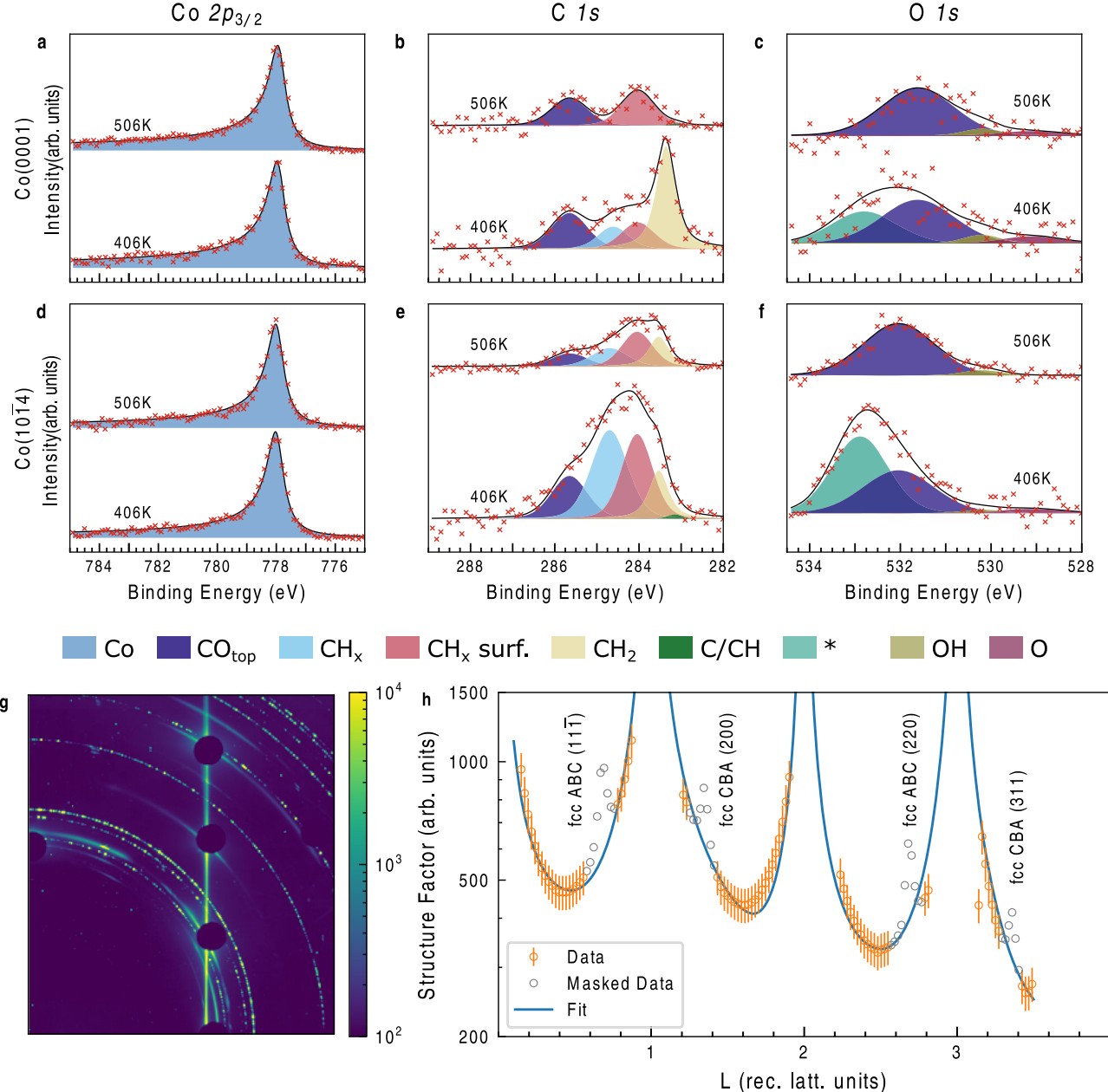

**Fig. 1 | 1 bar in-situ XPS and SXRD data under Fischer-Tropsch reaction conditions.** Surface state of Co(0001) sample at indicated temperatures in an atmosphere of CO:H₂ 1:2 at 1 bar studied by hard X-ray photoelectron spectroscopy (HAXPES) utilizing 4600 eV photons at 0.3° incidence. **a** shows Co $2p_{3/2}$ core-level spectra **b** displays the C $1s$ region and **c** depicts the O $1s$ region. Subplots **d, e, f** show the same conditions as **a, b, c** but for a Co(10$\bar{1}$4) surface. The columns of XPS data have constant scaling of the vertical axis. Subplot **g** shows a representative figure of

the high energy surface X-ray diffraction HESXRD data at 67.4 keV of the Co(0001) crystal (full set is shown in SI). The detector is protected by beam stops at the bulk Bragg peak positions. **h** X-ray structure factor extracted from the 2D diffraction data shown in **g** at reaction conditions at 496 K and 1 bar reaction mixture (1:2 CO:H₂), data from the hcp part of the surface used for the fit (orange circles with vertical lines indicating an estimation 10% relative error), data from the fcc part (grey circles), fit result (solid line). Source data are provided as a Source Data file.

The Co $2p_{3/2}$ spectrum is composed of a single peak at 778.0 eV that shows a completely metallic state with no shoulder at 780.0 eV[40] that would indicate an oxide. A carbide modification of Co would be expected at somewhat higher binding energy compared to the $Co^0$ metal peak if the shift follows as seen in Ni $2p_{3/2}$ upon carbon dissolution into bulk Ni[14] and the formation of Fe carbides[13]. The peaks in the C $1s$ region at 285.7 eV and in the O $1s$ region at 531.7 eV (blue) correspond to adsorbed CO in top position[29]. The C $1s$ feature at 284.1 eV at 506 K is related to hydrocarbon species and exemplifies the reaction intermediates. Other states are only observable at 406 K which will be further discussed below. A carbide species would be seen at around 283.0 eV and oxides at around 529.3 eV[41] and none are detected in the spectra.

The stepped Co(10$\bar{1}$4) surface is probed at the same conditions as the Co(0001) and the results there of are presented in Fig. 1d-f for the Co $2p_{3/2}$, C $1s$, and O $1s$ core-levels. Co is metallic during the reaction also on this facet. Adsorbates in the C $1s$ region, however, show a striking difference. Peaks at 284.1 eV and 284.7 eV are now observed at 406 K and as well at 506 K. Also, there is intensity in the region of 283.5 eV for both temperatures. Overall, the total C coverage is higher than for the flat surface. The O $1s$ intensity again shows only significant contributions of $CO_{top}$ adsorbates at 506 K and additionally intensity in the 533.0 eV region at 406 K. The origin of the 533.0 eV peak is still an open question. Since we observe no changes in the C $1s$ spectrum, which follow the variation of the 533.0 eV peak we can exclude an origin in oxygenated carbon-containing species. Chemisorbed water, either from contaminations in the purified gas or as a product of the FTS reaction, are other hypotheses which agree with the binding energy value of the peak. The known rapid desorption kinetics of water from Co surfaces—which has been observed even at temperatures as low as 170K[42]—decreases the likelihood of these hypotheses.

As XPS measurements are sensitive to the chemical state we have completed this data set with structure-sensitive high energy surface X-ray diffractometry (HESXRD) on a Co(0001) single crystal at 200 mbar and 456 K with 1:2 CO:$H_2$ under flow conditions (further conditions are shown in Supplementary Information S1). In Fig. 1g we display a maximum intensity per pixel from an angular scan rotating the sample around the surface normal in the range of the Co(0001) (1,0) crystal truncation rod (full experimental details are given in Methods). Our key observation is the appearance of a single surface rod at (1,0) reciprocal lattice units that indicates an unreconstructed hcp surface. A more detailed analysis shows that the behavior at partial pressures of 200 mbar and 1 bar of CO:$H_2$ 1:2 mixtures the surfaces do not reconstruct (see Supplementary Information S1), which is in line with previous findings of *operando* STM observations[4]. No indication for the formation of ordered carbide formation is found. In Fig. 1h we present the X-ray structure factor extracted from the data in **g**. From the fit we can deduce, that the surface is atomically smooth under all gas mixtures studied with a slight inward relaxation of the topmost layer of -0.04 Å. The fit improves by including CO molecules on the surface, but due to the small data set available, the occupancies and position could not be further refined. The full data set gives also evidence, that a few percent of the surface is fcc (111) terminated. Due to the low number of fcc-terminated sites the contribution from fcc can therefore be neglected for the XPS data analysis.

We can thereby conclude that on a Co(0001) single crystal at 1 bar and around 500 K during the operation of the FT reaction the Co surface remains fully metallic and retains an ordered, flat surface which exhibits considerable crystal truncation rod signal. Furthermore, there are no signs of a surface carbide or surface oxide indicating that the C $1s$ and O $1s$ spectral intensities are related to chemisorbed species.

## Detailed C $1s$ Spectral Interpretation

When inspecting the XPS spectra from the adsorbate at different conditions we have curve-fitted the data into specific components.

Since many different conditions in terms of pressure and temperature are measured an assigned spectroscopic component should at least be clearly visible as a peak or strong shoulder in one spectrum. The chemical assignment is based either on experimental spectra obtained from model compounds in ultrahigh vacuum (UHV) on Co(0001)[27,29,43,44] or on density functional theory (DFT) binding energy calculations (Supplementary Information S4). The binding energy scale potentially could differ by twotenths of an eV due to recoil effects at high kinetic energies that depends on the bonding strength (for discussion the reader is referred to Supplementary Information S2.d), however, we estimate these effects to be negligible.

Figures 2a and 2b shows the C $1s$ spectra from the reaction of CO and $H_2$ with a mix ratio of 1:2 at a pressure of 500 mbar and 200 mbar, respectively, at temperatures in the range of 406 K to 523 K over Co(0001). The corresponding O $1s$ spectra are shown in the Supplementary Information S2.a. We assign the 283.2 eV (green) feature to chemisorbed C or CH on the surface based on XPS spectra obtained from either decomposition of ethylene on Co(0001) as observed at 283.2 eV[43,44] or at 282.8 eV[29] and seen in exposure of CO and $H_2$ at 4 bar followed by evacuation at 283.3 eV[2]. The DFT calculation (Supplementary Information S4) gives a binding energy of 283.1 eV for adsorbed C, (energy scale corrected against experimental value of adsorbed CO in on-top position), whereas adsorbed CH has a somewhat lower value of 283.0 eV. With only such a small difference in C $1s$ binding energy between adsorbed C and CH and since there is a variation of the experimental value between 282.8 – 283.3 eV it is not possible to distinguish the two adsorbates we thus denote this peak C/CH at 283.2 eV (green).

The component observed at 283.5 eV (yellow) we assign to chemisorbed $CH_2$ species based on DFT calculations (see Supplementary Information S4). This binding energy has previously been reported as related to the $CH_3$ group in ethylidyne[29] but spectra of the chemisorbed ethylidyne molecule also include a peak corresponding C to the group bonded to the surface at 282.9 eV. Since we observe the 283.5 eV feature at several conditions without any low binding energy component the 283.5 eV peak cannot be associated with ethylidyne. With a similar argument, the 283.5 eV feature cannot be one of the carbons in adsorbed ethylene, where the adsorption site generates two inequivalent C atoms, since then it should be accompanied by a second carbon peak at 283.9 eV[29]. Next component, located at 284.1 eV (light red) we assign to hydrocarbon fragments, such as -$CH_2$- and -$CH_3$ groups on the surface based on DFT calculations and previous post-analysis experiments[2]. While some part of these hydrocarbon chains are in contact or in the proximity of the surface through undersaturated monomers, fully saturated parts are most likely pointing away from the surface and would correspond to the 284.7 eV (light blue) feature[45]. The energy difference between the initial and final states in a photoionization event is much smaller when the hydrocarbon group is directly bonded to the surface allowing for metallic screening of the core-hole state resulting in a lower binding energy for the parts of the hydrocarbon in direct contact with the surface (light red peak) than for the parts pointing away from the surface (light blue)[45]. Lastly, a clear peak originating from CO adsorbed in on-top configuration, denoted $CO_{top}$, is observed at 285.7 eV (dark blue). All these species are exemplarily depicted in the Supplementary Information in section S7. We find no indication of CO at other adsorption sites, such as the bridge or hollow sites, as commonly reported in UHV studies at liquid nitrogen temperatures[22] or in AP-XPS studies at ~1000x lower pressures[46], yet we observe only $CO_{top}$ in our experiments. The O $1s$ spectra contains mostly a feature associated with adsorbed CO in on-top position (see also Supplementary Information S2.a). No clear indication of any significant amount adsorbed O, OH, CHO, COH or $CH_3O$ species are detected on these single-crystal surfaces.

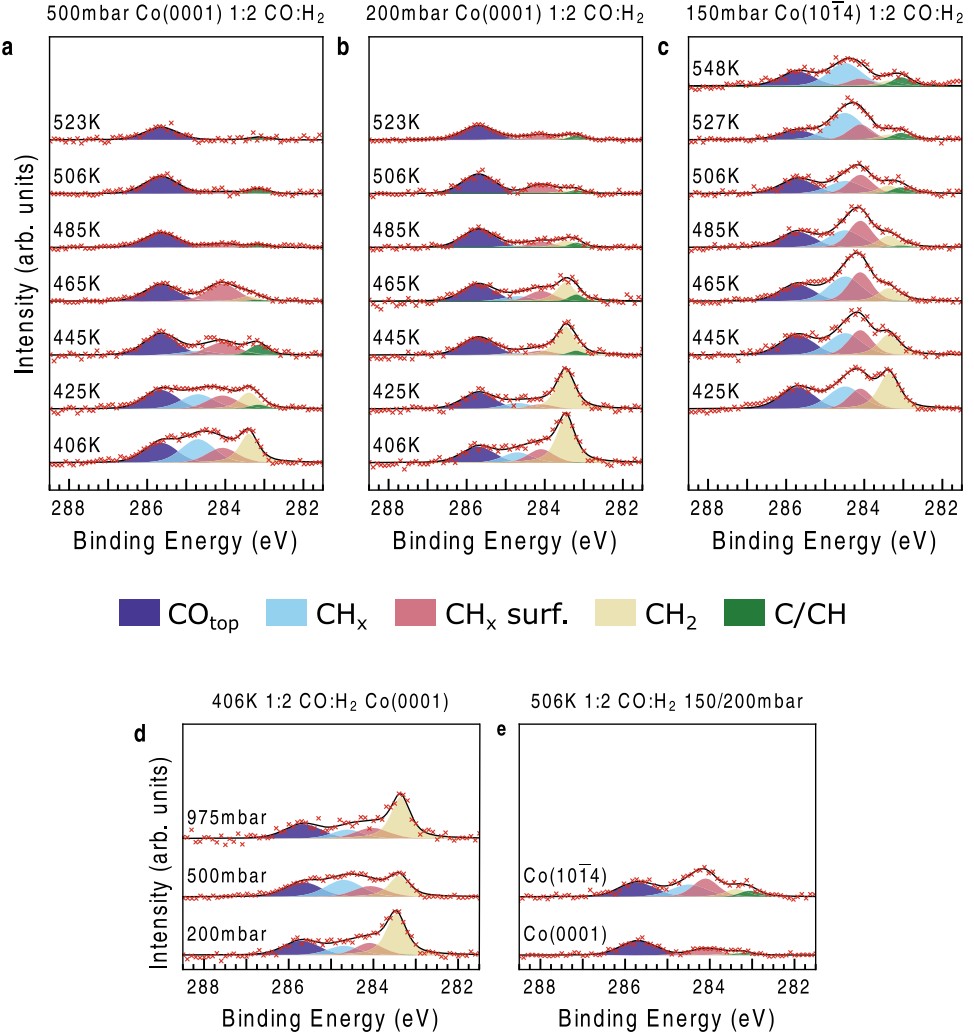

**Fig. 2 | Isobar C 1s spectra under reaction conditions.** We show Co(0001) at **a** 500 mbar and **b** 200 mbar. In **c** we show 150 mbar on Co(10$\bar{1}$4). All mixtures are 1:2 CO:H$_2$. Subplot **d** shows a comparison of 406 K data as function of pressure on the Co(0001) crystal, and subplot **e** displays a direct comparison of stepped Co(10$\bar{1}$4) and flat Co(0001) crystals at 406 K. All y-axes have the same scaling. Data has been normalized according to SI Section S6c. The color code is the same as in Fig. 1 **b/e**. Source data are provided as a Source Data file.

## Spectral Trends with Temperature, Pressure and Crystal Surface Orientation

Figure 2a shows the C 1s spectra at a pressure of 500 mbar from 406 K to 523 K on Co(0001). The coverage of the adsorbates has been determined through a specific normalization procedure (See Supplementary Information S2.c). We observe the largest total coverage of carbon containing species at the lowest temperature of 406 K corresponding to 1.5 ML with the -CH$_2$- peak (yellow) clearly dominating the spectrum, but also intensity is observed in the region of non-screened hydrocarbon chains (light blue). A "monolayer" is here defined relative to the surface atoms of the Co substrate. What is clearly noted is that the total coverage decreases with increasing temperature to below monolayer coverage for T > 480 K. We observe at 406 K a large amount of the hydrocarbon species with C atoms both bonded to the surface and with CH$_2$ and CH$_3$ groups away from the surface as well as surface bound CH$_2$ groups. As we reach the highest temperature of 523 K there is almost only CO on the surface and some small amount of adsorbed C/CH. Chain growth requires a considerable coverage of carbon species which are not fully saturated by hydrogen, which on the (0001) surface occurs below 485 K. This process can consequently occur at the lower temperature where there is a higher coverage of CH$_2$ groups and various adsorbed hydrocarbon species.

Figure 2b shows the same trend of temperatures but with a total pressure of 200 mbar on Co(0001). At the lowest temperature, we have an almost similar total coverage of 1.3 ML. We notice the same trend where the amount of species decreased with increasing temperature. What is mainly different is that the amount of CH$_2$-adsorbed species is now much higher in comparison to hydrocarbon molecules. The chain growth becomes less efficient with lower coverage. Again, the CO coverage ( ~ 1/3 ML) is almost independent of temperature.

Figure 2c shows the trend with the stepped Co(10$\bar{1}$4) surface at total pressure of 150 mbar. In general, we again observe an almost constant CO coverage but an increase in the hydrocarbon content and decrease of CH$_2$ adsorbed species indicating more efficient chain growth at steps compared to terraces. The total coverage at the lowest temperature of 425 K is 1.7 ML and compares well with our previous observations on the Co(0001) surface. The observation of coverages above 1 ML signifies that at these conditions we expect an amount of C on the surface able to cover more than all Co surface sites. The increase of atoms on the surface is due to the appearance of hydrocarbon chains linked to the surface but sticking out into the gas phase.

Figure 2d shows a comparison of spectra at different pressure on the Co(0001) surface at the FT temperature of 406 K. We observe an increase in the total coverage of adsorbed hydrocarbon species on the surface going to 1 bar, however, the relative distribution of different molecular fragments is somewhat similar at this low temperature. We can relate that the production of hydrocarbon at this temperature is

limited by the desorption of products and only becomes more efficient to a smaller degree with the increase in pressure since the surface is blocking its active sites due to kinetic hindrance.

Figure 2e shows a comparison of the C $1s$ spectra at the temperature of 506 K between the flat Co(0001) and the stepped Co(10$\bar{1}$4) surfaces. At this temperature, the catalyst is expected to be active for the FT reaction. There is a striking increase in the hydrocarbon content with the presence of steps. It is interesting to note that also the production of methane and minor hydrocarbon species increased by almost an order of magnitude between flat and stepped Co surfaces in a recent STM reactor study[3]. This increased reactivity was attributed to the lowering of the energy barriers for a rate limiting CO dissociation, and the increased hydrocarbon presence at all examined temperatures on the stepped crystal is fully consistent with this hypothesis[3].

Our data supports the view that CO dissociates most efficiently on the steps through a direct dissociation route and not the hydrogen-assisted mechanism. The direct route hypothesis is strengthened by not observing any CHO species on the surface that would be visible at 285.0 eV and 530.1 eV (O $1s$ spectra see: Supplementary Information S2.a). Although a weak component at the C $1s$ position could possibly be overlapping with adsorbed CO and hydrocarbon species, but there is no appreciable intensity at the low binding energy in the O $1s$ spectra[47]. Furthermore, ultrafast measurements using X-ray lasers have demonstrated that the CHO species could only exist in an extremely short lived transient regime with a life time of only a few picoseconds and could never build up any appreciable coverage during steady-state reaction conditions[47]. We do not observe any significant $CH_2O$ species with a calculated binding energy of 286.5 eV and 530.2 eV or $CH_3O$ species at 286.5 eV and 531.2 eV (O $1s$ spectra see: S2.a) pointing to that non-dissociated CO does not significantly contribute to the chain growth. Finally, as the coverage of adsorbed hydrocarbon species with more than two attached hydrogens per carbon is quite high the hydrogenation termination step leading to desorption would also be rate limiting. We therefore predict that both CO dissociation and the final hydrogenation leading to desorption is rate limiting under the current conditions on the stepped surface. On the flat surface the different hydrocarbon hydrogenation steps seem to be limiting as well, indicating an overall less active surface.

### Dynamics upon Changes in Reactant Composition

Figure 3 depicts the time dependence of the C $1s$ spectra related to the FT reaction of the Co(0001) surface by applying and removing CO while keeping a constant $H_2$ flow on the sample at 200 mbar total pressure. We performed experiments at 406 K (panel **a** with line extracts shown in **b**) and at 506 K (panel **c** with line extracts shown in **d**). At 406 K the sample surface is covered with a tiny amount of species at 284.5 eV initially. Upon exposure with CO there immediately appears intensity in the $CH_2$ and $CO_{top}$ regions (283.5 and 285.7 eV, respectively), indicating that some CO is dissociated and hydrogenated. Over an interval of approximately 30 min there is a continuous growth of the 284.5 eV state, indicating the appearance of longer chain hydrocarbons due to chain growth. Upon removal of the CO in the reaction mixture this component remains for a certain time while $CO_{top}$ and $CH_2$ are reacted away within the time resolution of this experiment. Continuing in this configuration, the 284.5 eV hydrocarbon peak intensity reduces, indicating facile reaction and departure from the surface aided by the presence of $H_2$, which supports our claim that a reaction is ongoing. We are here observing the rate limitation of the final hydrogenation step that removes the hydrocarbon species on the surface. A competing reaction to the hydrocarbon chain growth is the fast $CH_2$ hydrogenation into methane, which also explains the swift vanishing of the $CH_2$ contribution upon CO gas removal from the reaction mixture[27].

At 506 K, we observe mainly low $CO_{top}$ surface coverage and only to a negligible degree $C_xH_y$ species as compared to the 406 K experiment during reaction conditions, which is in line with the trends in the static measurements shown in Fig. 2. After CO is removed from the reaction mixture the intermediate hydrocarbons desorb or react away and the corresponding peak diminishes to baseline intensity. The dynamic response is much faster at the higher temperature. From these temperatures we derive that the surface is highly dynamic (i.e. turning over and in in-situ) and that changes in the conditions needs time to establish a steady state. Moreover, we observe that the rate limiting step changes from the formation of carbon chains at lower temperatures to the dissociation of CO at higher temperatures.

## Discussion

We have studied the two Co single crystal surfaces of (0001) and (10$\bar{1}$4) using in-situ XPS at almost 1000 times higher pressures than traditional NAPXPS and can directly probe adsorbates on the surface during the reaction. The C $1s$ and O $1s$ spectra shows only adsorbed species even at pressures close to 1 bar and the Co $2p_{3/2}$ spectra have no sign of an oxide or a carbide component. The surface X-ray diffraction results on Co(0001) demonstrate that the surface stays atomically smooth under reaction conditions. Thereby, there is no indication of any chemical or structural changes of the Co substrate surface region as the reaction proceeds. Our observations tip the scales in the discussion regarding the nature of the CO dissociation towards the direct or often denoted carbide mechanism since no sign of hydrogen-assisted dissociation in terms of CHO-detected species in the C or O $1s$ spectra. Furthermore, the chain growth involves only hydrocarbon species, since undissociated CO participation should show up as detected $CH_2O$ spectral components, which we did not observe. There are also no ethylidene or adsorbed ethylene intermediates detected pointing to simple chain growth of -$CH_x$- species resulting in an increasing amount of hydrocarbon species with groups both bonded directly to the surface but also pointing away towards the gas phase. Several of the current observations in terms of adsorbed CO and presence of hydrocarbon species on the surface have also been seen in the previous NAP study conducted at 100-1000 times lower pressures[46]. However, the higher pressure condition here also resulted in major differences as evidenced by the fact that CO adsorbs in the top site only, significantly smaller C/CH coverage, $C_xH_y$ species observed already at 406 K, no oxidation of the surface and no buildup of C at higher temperatures. We notice that the CO dissociation is more facile on the stepped Co single crystal surface. Due to the fast gas exchange rate in the virtual cell (See Supplementary Material section S2.f) accumulation of products in the gas phase and consequentially their reabsorption to the surface are of little significance. On the whole, the increased abundance of hydrocarbon species at 406 K on both surfaces, shows that a reaction is ongoing, yet the partly hydrogenated intermediates are not leaving the surface as rapidly as they are being produced. We thus argue that the final termination step in terms of hydrogenation (that is exchanging the carbon bond with a local C-H bond) is rate-limiting in this regime. At a temperature of 506 K the coverage of hydrocarbons on the surface is lower which indicates that the desorption of products is not limiting the reaction anymore. Since we observe an almost constant CO coverage in all our temperatures the availability of CO is also not a limit in this reaction. Thus, we infer that at this elevated temperature, the CO dissociation limits the rate of the reaction to a larger degree. It has been proposed that also the removal of oxygen atoms is a rate-limiting step[2,46] but in the current study no adsorbed oxygen was detected. We associate this lack of oxygen compared to the previous studies as most likely due to higher hydrogenation activity in in-situ studies at high pressures. Finally, our observation of the Co-based Fischer-Tropsch reaction is highly dynamic meaning that the involved species (despite a potentially long residence time) show changing adsorbate compositions as a direct consequence of changes in the reactant mixtures. The time for the delay is strongly temperature dependent and can be on the tens of minutes scale.

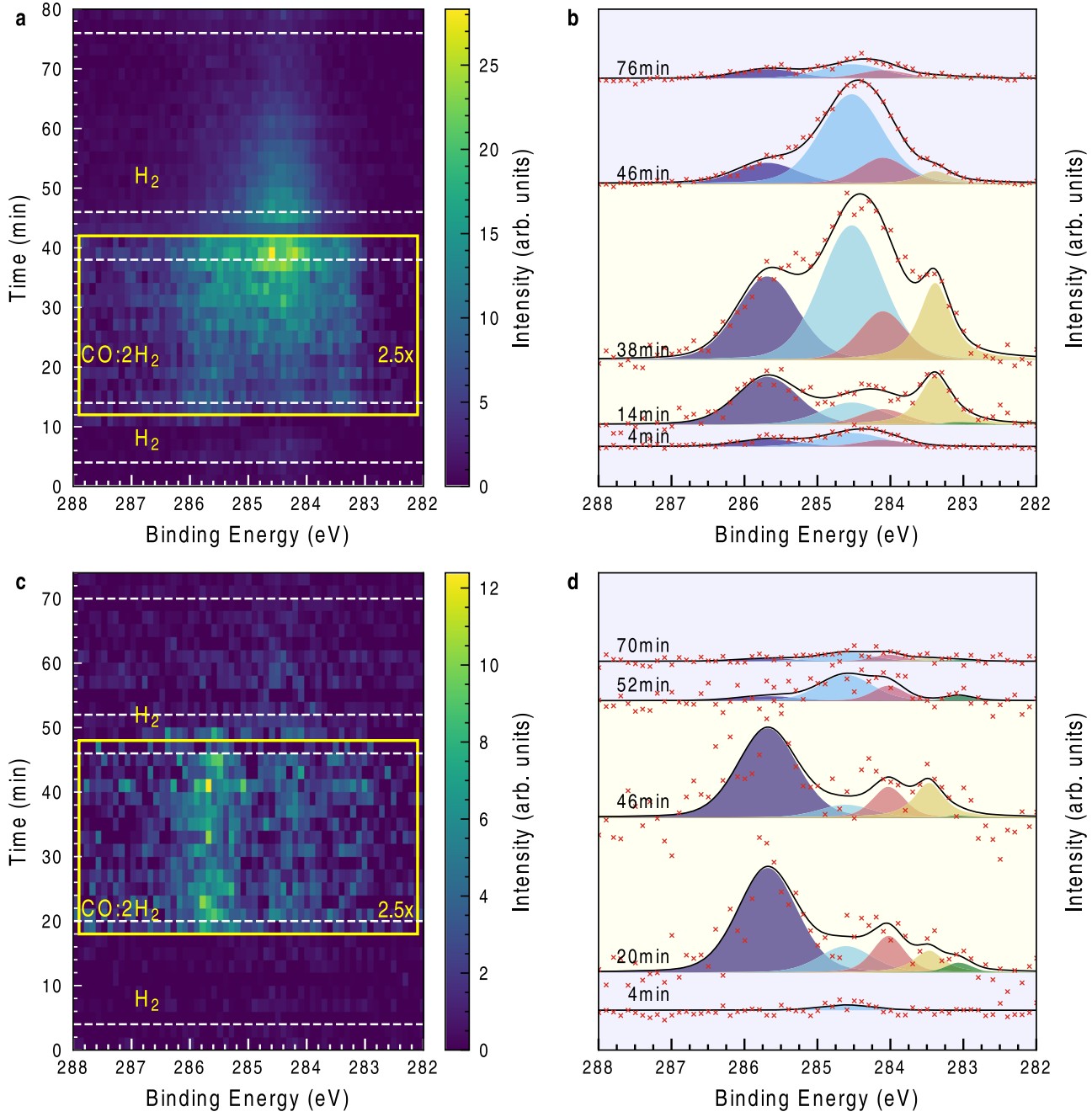

**Fig. 3 | Dynamic study of a CO addition and removal experiment. a** 2D time-resolved spectrum of the C $1s$ region at 406 K (**c** for 506 K) total pressure under CO flow is 200 mbar with a 1:2 CO:H$_2$ mixture. In the beginning and end the sample is subjected to a H$_2$ flow alone. The lines on the right are extracted from **a** and **b** at indicated times and resemble these reaction conditions from bottom to top: pure H$_2$, first 6 minutes under CO:H$_2$ mixture, last 6 min under CO:H$_2$ mixture, only H$_2$ after removal of CO from the mixture (red) and at the end of the experiment after at least 15 min in H$_2$(light blue). **b** & **d**: Spectra recorded at indicated times. Decomposition follows the Colo scheme of Fig. 1&2. Background: in hydrogen (cream), in 1:2 CO:H$_2$ (light blue). Source data are provided as a Source Data file.

## Methods

### Samples and gases

The experiments were performed on two hat-shaped cobalt crystals (N4.7) with the flat (0001) and the stepped (10$\bar{1}$4) direction exposed on the surface (R$_a$<30 nm, miscut <0.1°) with a top surface of 7 mm diameter (Surface Preparation Laboratory, SPL). It is worth noting that the stepped crystal exposes a high density of B5[36] sites that are regarded as an active site for the Co-based Fischer-Tropsch reaction[20]. The gases were obtained with a purity of 5 N for all gases except CO where the purity available could not exceed 4.7 N. For the X-ray photoelectron spectroscopy (XPS) experiments the gases were purified in the appropriate apparatus (SAES Getters/Entegris).

Under the measurements we recorded the temperature at the back of the sample with a type N thermocouple. Sample front temperature has been calibrated under various gas loads for a comparable specimen with a thermocouple spot-welded to the front. We expect that our temperature measurements agree to be better than $\Delta T < 15$K.

### Polaris XPS experimental setup

The POLARIS setup is placed at beamline P22, DESY. The spectrometer utilizes a virtual cell approach, where a gas stream is directed onto a flat specimen and X-rays reach the interaction volume under grazing incidence conditions of 0.3°[31]. In this way a radially symmetric local

**Table 1 | Peak Model for the quantification of C $1s$ peaks**

| Name | Center (eV) | Gaussian FWHM (eV) | Lorentzian FWHM (eV) | Type |
|---|---|---|---|---|
| C | 283.00..283.20 | 0.30..0.45 | 0.20..0.50 | Pseudo-Voigt[52] |
| $CH_2$ | 283.3..283.55 | 0.5..0.8 | 0.2..0.3 | Pseudo-Voigt[53] |
| $CH_{surf}$ | 283.80..284.10 | 0.20..1.00 | 0.00..0.50 | Pseudo-Voigt |
| CH | 284.00..284.70 | 0.75..0.90 | 0.00..0.50 | Pseudo-Voigt |
| COtop | 285.65..285.85 | 0.20..1.50 | 0.00..0.50 | Pseudo-Voigt |

**Table 2 | Peak Model for the quantification of O $1s$ peaks**

| Name | Center (eV) | Gaussian FWHM (eV) | Lorentzian FWHM (eV) | Type |
|---|---|---|---|---|
| O | 529.10..529.30 | 1.20..1.40 | 0.15 | Pseudo-Voigt |
| OH | 529.80..530.20 | 0.40..2.00 | 0.15 | Pseudo-Voigt |
| COtop | 531.60..532.20 | 1.70..1.90 | 0.15 | Pseudo-Voigt |
| * | 532.80..533.20 | 0.90..1.60 | 0.15 | Pseudo-Voigt |

**Table 3 | Peak Model for the quantification of the Co $2p_{3/2}$ peak**

| Name | Center (eV) | Lorentzian FWHM (eV) | Asymmetry (a.u.) | Type |
|---|---|---|---|---|
| Co Metal | 777.5..778.3 | 0.0..1.0 | 0.18..0.40 | Doniach- Sunjic[53] |

high-pressure cushion is formed (d ~ 2 mm), while the rest of the chamber experiences much lower pressures (i.e. when the pressure in the probed volume is ~1000 mbar the chamber pressure is 10 mbar). The excited electrons are collected at distances of approximately 30 µm by a line array of circular apertures, well matching the stretched-out X-ray footprint due to the grazing incidence geometry.

For the XPS experiment we used a double bounce mono-chromator with Si(311) crystals tuning the photon energy to 4600 eV. The electron analyzer was used with an 800 µm curved slit and a pass energy of 100 eV. The total energy resolution where the expected photon energy bandwidth and the electron analyser resolution is folded together account for $\Delta E_{FWMH} \leq 300$ meV. The beamline optics use a horizontally bent elliptical mirror and a vertical cylindrical focusing mirror to achieve a beam footprint on the order of $15 \times 15 \mu m^2$, which was measured with a polished YAG crystal at regular intervals during the experiment.

Due to the specific design of this experimental setup (i.e. an outward-flowing gas jet), no contamination from the heater or chamber can reach the sample surface when gas flow is applied. The pressure in the virtual pressure cell was estimated from a calibration done by a previously used method[31] (See Supplementary Information S5).

All XPS spectra were acquired in an add-dimension mode indicating that a list of short C $1s$, O $1s$, Co $2p_{3/2}$ spectra are recorded repeatedly and summed together for statistics. This method allows to distribute slow surface changes under the reactions into all three spectra in similar weights such that distortions between spectra are expected to be negligible and developments can be traced quasi simultaneously in several core-levels.

## XPS data processing

All spectra are recorded with a binding energy calibration to the Fermi edge of a clean metal. A correction for the adsorbate binding energies due to the recoil effect has been neglected (see Supplementary Information S2d). The spectra were first converted to counts per second and were plotted in each panel with offsets for visibility. For comparison, we normalized the spectra as described in Supplementary Information S2c. We observe 4 distinct peaks in O $1s$ spectra. These are observed at 532.5 eV, 531.7 eV, 530.2 eV, and 529.1 eV. The first peak is unassigned and the second peak assigned to $CO_{top}$, whereas the last two are tentatively assigned to chemisorbed O and OH states, which,

however, are not observed in an appreciable quantity. In the C $1s$ spectra we notice 5 distinct features observed at 285.7 eV, 284.7 eV, 284.1 eV, 283.5 eV and 283.2 eV.

The quantification of the peaks is performed by applying a global fit model that finds a maximum likelihood optimization using the Levenberg-Marquardt algorithm of the peak positions and shapes for all spectra of a set. The constraints of the fitting parameters are given in Table 1 (for the C $1s$ region), Table 2 (for the O $1s$ region) and Table 3 (for the Co 2p region). The background estimation has been performed by subtracting a Shirley function where the Shirley parameter is a free variable in every spectrum[48].

## SXRD experimental setup

The surface X-ray diffraction (SXRD) experiment[49] was performed at beamline P21.2 at DESY using a beam with ~3×10 µm (VxH) FWHM, an energy of 67.4 keV and a glancing angle of 0.05°[50,51]. The detector (VAREX XRD 4343CT, 150 µm pixel size) was placed 1.4 m away from the interaction zone. The sample temperature was controlled using a BN-encapsulated graphite heater. The gases of 5 N purity for Ar, and $H_2$ and 4.7 N purity CO (same purities as in the XPS experiment) were delivered to an X-ray transparent Be dome and therein directed onto the same Co(0001) single crystal as for the XPS experiment. The CO gas was additionally purified from $Ni(CO)_4$ using a copper carbonyl trap. The experimental setup is in details described elsewhere[49]. Scans were taken over a range of 105°.

## Sample cleaning

Prior to the reaction studies the Co(0001) and Co(10$\bar{1}$4 ) samples were cleaned by established procedures of repeated sputtering (1 keV, ~1 µA, 30 min) and annealing (523 K, 30 min) processes in ultra-high vacuum. In the XPS experiments the single crystals were also chemically cleaned by dosing of 0.16 liter per minute $O_2$ for 5 s followed by annealing in 1 liter per minute $H_2$ at a sample distance of 30 µm and 473 K. In between each measurement the samples were kept in $H_2$ flows greater or equal 0.16lpm (p < 50 mbar, negligible scattering in gas) to limit contamination from the vacuum environment, while we evaluated the surface cleanliness of C $1s$, O $1s$, S $1s$, Si $1s$ containing species. The intensity sum of these was kept below 5at% of the Co $2p_{3/2}$ signal. The Si $1s$ peak is orders of magnitude more sensitive to hard X-rays (by about a factor 100x) than both Si 2p core-levels combined.

## Data availability
Experimental data were generated at the PETRA III facility at the DESY Research Centre of the Helmholtz Association. Source data are provided with this paper.

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

## Acknowledgements

The work was supported by the Swedish Research Council (Vetenskapsrådet, VR, project 2017-00559 and project 2013-8823), the Knut & Alice Wallenberg (KAW, grant nr. 2016.0042) foundation, as well as the Swedish Foundation for strategic research (Stiftelsen för Strategisk Forskning, SSF, Proj. ITM 17-0034). The research leading to this result has also been supported by the project CALIPSO plus under Grant Agreement 730872 from the EU Framework Program for Research and Innovation HORIZON 2020. The experimental part of this research was carried out at P22 and P21 beamlines at DESY, a member of the Helmholtz Association (HGF). Beamtime was given for in house research proposals. The DFT calculations were performed using resources provided by the Swedish National Infrastructure for Computing (SNIC) at the HPC2N center. The authors would like to acknowledge the help of the P22 beamline engineer Katrin Ederer, and the Technical Division at Stockholm University.

## Author contributions

P.L. with input from A.N. planned the experiments at Petra III. P.L., D.D., Mi.Sh., Ma.So, F.G.M., C.M.G., J.G., H.-Y.W., A.H., R.R., A.S., C.S., A.N., P.A. participated in the XPS experimental work, while Z.H., A.S., H.N., V.V., J.-C.S., R.G., S.T. and L.J. participated in the SXRD measurements. P.L. extracted and plotted the SXRD data and A.S. fitted it. G.L.S.R. performed theoretical calculations. P.L. and Mi.Sh developed the data analysis software and P.L. did the data analysis. P.L. and AN wrote the manuscript. All authors contributed to the literature research, result discussion and manuscript improvement.

## Funding

## Competing interests

The authors declare no competing interests.
