## [Transparent Peer Review file · Nature Communications]

In-situ Probing of the Fischer-Tropsch Reaction on Co Single Crystal Surfaces up to 1 bar

Corresponding Author: Professor Anders Nilsson

Version 0:

Reviewer comments:

Reviewer #1

(Remarks to the Author)

The authors report on the use of high-pressure XPS, making use of a unique research instrument, thereby illustrating its power/potential of assessing the state of cobalt (and oxygen and carbon) as well as the different surface adducts formed during the industrially important Fischer-Tropsch synthesis process. The use of this equipment as well as the data for an industrially valuable catalyst system is a clear novelty and therefore the work is in principle publishable in a multidisciplinary journal such as Nature Communications. The authors have carefully analyzed the data and based on this they note that (a) the cobalt remains in its metallic state; and (b) the surface adducts rule out one of the possible routes for the formation of longer hydrocarbons from CO and hydrogen (no CHO species observed). The authors report not only XPS but also surface-sensitive XRD; although the two surfaces assessed with XPS, are not both assessed by XRD; that is something that is not ideal and also the fact that the B5s sites are preferentially found on the stepped surfaces, and their - proposed - importance is not that well discussed. Overall, I believe the work would benefit from a more "symmetric" approach in using the two methods; and secondly, the authors have to realize that their surfaces are crystals which are "ideal" but not that realistic in terms of a supported Co metal nanoparticle on an oxidic support. Within this context, one could argue where the "claims" for the presence of oxidic or carbidic cobalt, as well as the presence of CHO type species are coming from. If they do not exist on a planar/ideal surface, then they may originate from the interface between a small cobalt nanoparticle and the support oxide. As this is not researched in this work, one could argue that the claims made should be also rephrased as this is then the "boundary" conditions why it was not observed or noted in the spectra obtained with XPS and XRD. AS a side remark, I feel that the discussion on the chemistry of cobalt Fischer-Tropsch synthesis could be improved (taking into account more/other reference studies). Another point of attention is that I note that the assignments are based on sometimes rather weak bands; and on deconvolution of spectra which need some more statistical validation. A final comment is on the use of the work operando; it is not fully clear but as I understand there is no activity measurements hence the word in situ is a better one to define the experiments described in the paper. Summarizing, this is a very nice and original study on an important catalytic material, which deserves publication, but some revisions/additions have to be considered before the paper can be published in Nature Communications.

Reviewer #2

(Remarks to the Author)

Lömker et al. present a study on the Co-catalyzed Fischer-Tropsch synthesis. The Fischer-Tropsch synthesis is one of the best-studied catalytic reactions, but this study is unique as the surface of a Co single crystal could be investigated under conditions, pressures up to 1 bar and elevated temperatures, where catalytic turnover is known to take place. A newly developed XPS technique has been used that can operate at 1 bar (previous versions, often termed near-ambient pressure XPS, were restricted to pressures up to ~10 mbar) and combined it with surface X-ray diffraction under the same conditions. It is shown that the Co surface is not covered by a carbide or an oxide but remains metallic and has the hcp structure of bulk Co. This work is a major achievement as it provides insight into the conditions of the surface of an operating catalyst. In my view, the ms may qualify for Nature Communications, but there are several issues that should be clarified before I can recommend publication.

/1/ What is the emission depths of the photoelectrons that are excited by the 4600 eV photons used here? For the surface species, the emission depth is irrelevant but it may play a role for the interpretation of the Co-related peaks. These could be overwhelmed by the bulk of the crystal.

/2/ I have doubts about the interpretation of the O 1s peak at 533 eV as adsorbed H₂O (Figs. 1c and f, 406 K). Of course,

water is a product of the Fischer-Tropsch reaction, but published TPD data show that H₂O has fully desorbed from a Co(0001) surface already at 170 K (Weststrate et al., J. Phys. Chem. C, 127, 2974, 2023). It is hard to see how H₂O can accumulate on the surface at 406 K.

/3/ The light grey original data points in all XP spectra are hardly visible. In this form the quality of the peak fits cannot be assessed.

/4/ It is not clear how the small fcc fraction of the Co lattice was evaluated from the diffraction data. Please explain.

/5/ The O 1s and C 1s signals of gas phase CO are outside the indicated energy ranges. Just for completeness, have the gas phase signals been measured?

/6/ The introduction, namely lines 49/50 and 59-62, creates the impression that there has been no prior knowledge at all about the chemical state of the Co surface at a syngas pressure of 1 bar. It is clear that STM, one of the previously used methods under reaction conditions, has no direct chemical sensitivity. However, one can often conclude about the surface chemistry from the observed structures. For example, in an operando STM study on the CO oxidation on a Pt(110) surface (Hendriksen et al., Top. Catal. 36, 43, 2005) it was shown that the surface became rough during the reaction which could be interpreted as the formation of a surface oxide. On the Co(0001) surface, Co carbide would probably have been identified by its characteristic structure, so that its absence indicated that the surface was metallic (refs. 3, 4, 25, and Banerjee et al., J. Phys. Chem. Lett. 7, 1996, 2016). The achievements of the present study are not weakened by these facts.

/7/ The introduction, lines 77-79, also suggests that infrared spectroscopy under reaction conditions, ~1 bar syngas, was unable to detect anything else than adsorbed CO (based on ref. 24). However, there is a DRIFT study on an oxide-supported Co catalyst at 1 bar, Paredes-Nunez et al., Catal. Today, 242, 178, 2015, presenting spectra that show hydrocarbons and other organic compounds in addition to the CO. These species were probably not adsorbed on the Co but on the oxide support. The fact that they were not seen in work on Co single crystals in ref. 24 (also in Beitel et al., J. Phys. Chem. B, 101, 4035, 1997) could therefore just be explained by the fact that no such species were present. It must not be a limitation of infrared spectroscopy. In fact, the conditions in the Beitel paper, 300 mbar syngas, 490 K, were similar to those in the experiment of Fig. 2b, 200 mbar, 485 K, of the present paper. According to the XPS, mainly adsorbed CO is present but hardly any hydrocarbons.

/8/ From S8, it is not clear which physical system was used in the DFT calculations. A layer of molecules without the metal surface?

/9/ Minor corrections: Lines 54 - 56: In ref. 12 Co₂C was investigated, not CoC₂. Lines 65 - 67: "Afterwards such C can either attach to other carbon atoms and thus grow the hydrocarbon chain or ... "; a coupling reaction between C atoms would give chains only consisting of C atoms, not a hydrocarbon chain. Lines 96 - 98: This sentence seems incomplete. Lines 106 - 109: The stepped crystal investigated in ref. 3 was Co(1.0.-1.15), not Co(1.0.-1.5). Lines 247/248: Fully hydrogen saturated species would be methane, ethane, etc.; the adsorbed hydrocarbon fragments cannot be fully hydrogen-saturated.

Reviewer #3

(Remarks to the Author)

In the submission the authors report an operando X-ray photoelectron spectroscopy study of Co(0001) and Co(10 $\bar{1}$ 4) and operando high energy surface X-ray diffraction of Co(0001) during the Fischer-Tropsch reaction at 0.15 bar - 1 bar and 406 K - 548 K in a H₂/CO gas mixture. Some interesting results were observed, such as the atomically smooth metallic Co surfaces under all conditions and the coverage of chemisorbed species ranges from 0.4 – 1.7 monolayers depending on pressure and temperature, a rate-limiting direct CO dissociation pathway with an exclusive participation of hydrocarbon species in the chain growth. These results deepen the fundamental understanding of Co-catalyzed FT synthesis of broad interest and great importance. The manuscript can be accepted after the authors clarify the following issues:

1. Fundamental studies of Co-catalyzed FT reaction were carried out rather extensively under UHV and NAP conditions. Although the submission report the results under an elevated pressure, most results are similar to those reported previously (such as Ref. 40), and a main contribution of the present work is to demonstrate that the previous results also hold under reaction conditions approaching the working condition more closely. The authors are strongly suggested to discuss their results in the context of previous work appropriately by clarifying this fact.
2. It is not appropriate to use the term "operando" because no reaction data together with the in situ characterizations is provided. Were there any gaseous products observed by XPS?
3. The arguments of direct CO dissociation and termination step as the rate-limiting steps need to be re-considered. Under the adopted reaction conditions, the reaction seems to be in an equilibrium state which can not be used to justify the rate-limiting step. In the Ref. 2 and 40, the removal of oxygen adatoms resulted from CO dissociation is identified as the rate-limiting step by studies from temperatures starting at RT.
4. What is the definition of monolayer? Please clarify the speciation of adsorbates with the total coverages above 1 ML.
5. About the assignments of C 1s XPS peaks, the results of well-defined alkyl fragments on Co single crystal surfaces (J. Phys. Chem. C 2019, 123, 7740–7748; J. Phys. Chem. C 2020, 124, 24786–24794; Surface Science and Technology (2023) 1: 5

) can be cited.

Version 1:

Reviewer comments:

Reviewer #2

(Remarks to the Author)

Report on revised ms "Operando Probing of the Fischer-Tropsch Reaction ..." by Lomker et al.

In essence, I am content with the revision of this ms and with the comments to my report. However, I still doubt the interpretation of the O 1s peak at 533 eV as adsorbed H₂O. The authors argue that the presence of H₂O on the Co surface at the high temperature of 406 K can be understood by the reaction, namely by the relatively high H₂O production vs. desorption rate. In my view, this argument is not valid. The production rate of H₂O is identical to the TOF of the Fischer-Tropsch synthesis, which is a well-known quantity. Values given in the review article by Ribeiro et al., Catal. Rev.- Sci. Eng. 39, 49 (1997) vary between 10⁽⁻³⁾ and 10⁽⁻¹⁾ s⁽⁻¹⁾, depending on the sample and experimental conditions. The desorption rate of H₂O can be estimated from the H₂O adsorption energy on Co(0001) for which Weststrate et al., J. Phys. Chem. C, 127, 2974 (2023), measured a value of 57 kJ mol⁽⁻¹⁾. Application of the Arrhenius equation and using a preexponential factor of 10⁽¹³⁾ s⁽⁻¹⁾ gives a desorption rate at 406 K of approximately 5 x 10⁽⁵⁾. This value is more than 6 orders of magnitude higher than the production rate, showing that, at 406 K, H₂O cannot exist on the metallic Co in measurable quantities. The authors cite J. Phys. Chem. C, 114, 17029, 2010 (fig. 8) as an example where H₂O desorption from a Co sample was reported to occur at temperatures as high as 400 K. However, the authors of this paper showed that in this experiment H₂O desorbed from an oxide surface rather than from metallic Co, and that H₂O was strongly bound on the oxide surface. I recommend that the authors mark the 533 eV peak as unexplained; it does not compromise the message of the paper.

Minor: In the revised version, the authors provide information about the DFT calculations, but they do not say whether these were done spin-polarized (cobalt is magnetic). The sentence "Near-ambient XPS (NAPXPS) studies ..." is still grammatically incorrect.

Reviewer #3

(Remarks to the Author)

In the revised submission, the authors replied appropriately to most of my previous comments except that the authors still do not consider the adsorption-desorption equilibrium of products on the observed C-containing species in the in situ XPS spectra at low temperatures. The presence of such processes is demonstrated by the observation of adsorbed H₂O species at low temperatures. The neglect of such processes also leads to a puzzling argument that the rate limiting step changes from the formation of carbon chains at lower temperatures to the dissociation of CO at higher temperatures (Lines 382-383). Since the formation of carbon chains occurs after the CO dissociation, I can not see how the rate limiting step changes from the formation of carbon chains at lower temperatures to the dissociation of CO at higher temperatures without considering the facilitated desorption process at high temperatures. After this issue is clarified, the manuscript can be accepted.

Version 2:

Reviewer comments:

Reviewer #2

(Remarks to the Author)

The modifications of the ms following my report on the previous version are all appropriate. I can recommend publication of this article without further changes.

Reviewer #3

(Remarks to the Author)

Principally I support the publication of the manuscript. But I would like to encourage the authors to comment on which final termination steps in terms of hydrogenation would overcome barriers higher or as high as CO dissociation in the manuscript, because this finding will be very interesting, and as far as I know, has not been proposed before.

RESPONSE TO REVIEWERS' COMMENTS

Reviewer 1:

We would like to thank the reviewer for taking the time to read the manuscript and provide a critical review. The comments have helped us to develop a stronger and clearer manuscript.

The authors report on the use of high-pressure XPS, making use of a unique research instrument, thereby illustrating its power/potential of assessing the state of cobalt (and oxygen and carbon) as well as the different surface adducts formed during the industrially important Fischer-Tropsch synthesis process. The use of this equipment as well as the data for an industrially valuable catalyst system is a clear novelty and therefore the work is in principle publishable in a multidisciplinary journal such as Nature Communications. The authors have carefully analyzed the data and based on this they note that (a) the cobalt remains in its metallic state; and (b) the surface adducts rule out one of the possible routes for the formation of longer hydrocarbons from CO and hydrogen (no CHO species observed).

The authors report not only XPS but also surface-sensitive XRD; although the two surfaces assessed with XPS, are not both assessed by XRD; that is something that is not ideal and also the fact that the B5s sites are preferentially found on the stepped surfaces, and their - proposed - importance is not that well discussed. Overall, I believe the work would benefit from a more "symmetric" approach in using the two methods;

Author response:

We agree with the reviewer that it would be ideal to include SXRD of both the Co(0001) and Co(10 $\bar{1}$ 4) but, unfortunately, the allocated beamtime from DESY for SXRD only allowed us to study one surface. Since we detected metallic Co in both XPS and SXRD on Co(0001) and metallic Co with XPS on Co(10 $\bar{1}$ 4), we have no reason to suspect that SXRD would also show evidence of non-metallic Co. Naturally, it would be most interesting for a future study of Co(10 $\bar{1}$ 4) with SXRD to follow for instance step migration under reaction conditions.

The proposed discussion regarding the B5 sites is a great suggestion which we are happy to include in our manuscript. We have added additional signifying statements about the B5 sites in the main text and we thank the referee for making this point:

“In particular, the size dependent effects that show high activity for certain size nanoparticles have in the 2010s been shown to be linked to the relative abundance of B5 sites that appear at the intersection of threefold and fourfold coordinated sites^{33,35–38}. These sites can be found on the Co(10 $\bar{1}$ 4) stepped surface and thus further the explanation why stepped surfaces are observed with higher activities (See SI S11 and Supplementary Figure 9).”

Further we have added Section S11 and Figure S9 in the SI to illustrate the specific site better:

In this particular surface cut the planes expose differently large terraces that can be 1-2 atoms long and exhibit symmetries of alternating planes consisting of 3-fold coordinated Co atoms and 4-fold coordinated Co atoms. Each adjacent site of a 4-fold plane that is below a 3-fold plane exposes a B5 type site where an adsorbed atom would be in the proximity of 5 surrounding neighbors (4 from the same plane and 1 from the 3-fold plane above this plane). This particular geometry facilitates an enhanced catalytic activity of the catalyst^{S31} and is not existing in the flat Co(0001) surface, as its maximum coordination number is 4. A view onto this surface is presented in Supplementary Figure 9.

Supplementary Figure 1. **B5 Sites on Co(10 $\bar{1}$ 4).** Co(10 $\bar{1}$ 4) surface displayed from above, visualized with SurfaceExplorer. The intersection of the lower 4-fold coordinated terrace with the above-lying 3-fold coordinated terrace creates a unique 5-fold coordinated site that is denoted as B5 site and is connected to higher activity of this Co surface.

and secondly, the authors have to realize that their surfaces are crystals which are "ideal" but not that realistic in terms of a supported Co metal nanoparticle on an oxidic support. Within this context, one could argue where the "claims" for the presence of oxidic or carbidic cobalt, as well as the presence of CHO type species are coming from. If they do not exist on a planar/ideal surface, then they may originate from the interface between a small cobalt nanoparticle and the support oxide. As this is not researched in this work, one could argue that the claims made should be also rephrased as this is then the "boundary" conditions why it was not observed or noted in the spectra obtained with XPS and XRD.

Author response:

We are well aware of the ideal nature of single crystal catalysts and the limited scope of this work was already in the original manuscript clearly stated in both title and abstract. It is possible that a CHO type species might exist on oxide supported catalysts, but it is beyond the current work to discuss systems that we have no information about. Of course, extension of the current work with Co nanoparticles on oxide would be most interesting to conduct in the future. In the introduction we have reworded the sentence after the enumeration of the two single crystal surfaces such that it now reads

"We find that these Co surfaces remain..."

We have changed the last sentence of Section 2.b to now read:

"No clear indication of any significant amount of adsorbed O, OH, CHO, COH or CH₃O species are detected on these single crystal surfaces."

As a side remark, I feel that the discussion on the chemistry of cobalt Fischer-Tropsch synthesis could be improved (taking into account more/other reference studies).

Author response:

We have added some more text in the first paragraph of the introduction about the chemistry of the Fischer-Tropsch reaction. However, we apologize that this is a rather short introduction because we are constrained by the space limit of Nature Communications, particularly as we also add text based on other reviewer's suggestions.

"Presently, the Fischer-Tropsch reaction utilizes Fe, Ru or Co based catalysts that yield different hydrocarbon distributions (i.e., with regard to the abundance of shorter or longer C-chains in the product stream). Depending on the material the reactions follow these main pathways:

Further, the created water can react with the CO by the water gas shift reaction:

The latter reaction creates more H₂ at the expense of CO, but on Co catalysts it is not of significant importance thus the gas mixture of 1:2 CO:H₂ is commonly employed.¹

Further, as an answer to another referee request, we have intensified the discussion with existing literature (See Reviewer 3, Point 1 & 3).

Another point of attention is that I note that the assignments are based on sometimes rather weak bands; and on deconvolution of spectra which need some more statistical validation.

The reason why some of the assignments seem weak is that a global fit model is employed. What it means is that the same peaks are used to model all spectra with the same width, shape and binding energy; only the amplitude of the components varies between spectra. Consequently, if a component is clearly visible in the raw data at one of the examined conditions, we have fitted this component for the entire series, even if the intensity might be very low in other spectra. A very low intensity should be interpreted as nothing more, nothing less than a very low (and potentially insignificant) coverage. This approach was already described in the SI but now is described in the first Section of the Results and Discussion:

“In this work we apply the following procedure to justify the application of a peak in the spectra: The peak needs to be clearly visible in at least one instance of our measurements to be appended to the peak model. These individual observations we compile into a global fit model which we use for all spectra. We apply this global fit model to each series of spectra such that the defining peak shapes are common to all spectra as described below.”

A final comment is on the use of the work operando; it is not fully clear but as I understand there is no activity measurements hence the word in situ is a better one to define the experiments described in the paper.

Author response:

We have rephrased our findings to reflect the in-situ nature of our observation. All claims to the *operando* nature of our findings have been transcribed to *in-situ* including in the title. The title now reads:

“In-situ Probing of the Fischer-Tropsch Reaction on Co Single Crystal Surfaces up to 1 bar”

We have rephrased the second to last sentence of the Introduction to read:

*“Furthermore, we show the facile appearance and disappearance of C_xH_y adsorbates as seen in section II.d as indicators of a state **where the reaction should occur.**”*

Also, the third to last sentence in the paragraph describing Fig. 3 has been modified to read:

*“Continuing in this configuration, the 284.5eV hydrocarbon peak intensity reduces, indicating facile reaction and departure from the surface aided by the presence of H₂, which supports our **claim that a reaction is ongoing.**”*

Summarizing, this is a very nice and original study on an important catalytic material, which deserves publication, but some revisions/additions have to be considered before the paper can be published in Nature Communications.

Author response:

We thank the reviewer for this comment.

Reviewer 2:

Lömker et al. present a study on the Co-catalyzed Fischer-Tropsch synthesis. The Fischer-Tropsch synthesis is one of the best-studied catalytic reactions, but this study is unique as the surface of a Co single crystal could be investigated under conditions, pressures up to 1 bar and elevated temperatures, where catalytic turnover is known to take place. A newly developed XPS technique has been used that can operate at 1 bar (previous versions, often termed near-ambient pressure XPS, were restricted to pressures up to ~10 mbar) and combined it with surface X-ray diffraction under the same conditions. It is shown that the Co surface is not covered by a carbide or an oxide but remains metallic and has the hcp structure of bulk Co. This work is a major achievement as it provides insight into the conditions of the surface of an operating catalyst. In my view, the ms may qualify for Nature Communications, but there are several issues that should be clarified before I can recommend publication.

/1/ What is the emission depths of the photoelectrons that are excited by the 4600 eV photons used here? For the surface species, the emission depth is irrelevant but it may play a role for the interpretation of the Co-related peaks. These could be overwhelmed by the bulk of the crystal.

Author response:

The reviewer poses a great question, and we agree that it warrants further clarification. In our case, the probing depth is not only reliant on the emission depth of the photoelectrons but also the penetration depth of the X-rays. We have added the following clarification into the main text:

“The combined effect of X-ray penetration depth and electron inelastic mean free path yields an effective inelastic mean free path comparable with laboratory XPS systems of about 1.4nm at the C 1s core-level and 1.3nm at the Co 2p core-level¹³.”

In our other works (on Fe: 10.1021/acscatal.2c00905 & on Ni: 10.1021/acs.jpcc.2c07650) we see clear signals of surface carburization that are absent in the data we present here, which corroborates further the surface sensitivity of this setup.

/2/ I have doubts about the interpretation of the O 1s peak at 533 eV as adsorbed H₂O (Figs. 1c and f, 406 K). Of course, water is a product of the Fischer-Tropsch reaction, but published TPD data show that H₂O has fully desorbed from a Co(0001) surface already at 170 K (Weststrate et al., J. Phys. Chem. C, 127, 2974, 2023). It is hard to see how H₂O can accumulate on the surface at 406 K.

Author response:

We understand that this might be surprising and it was for us. However, the presence of water is not through adsorption from the gas phase as in UHV studies but instead the outcome of the reaction. The coverage is determined by the residence time and the production rate. Water can therefore build up a higher coverage compared to TPD-measurements. Furthermore, there are studies with high coverage of water indicating desorption maximum as high as of 400 K, see Figure 8 in J. Phys. Chem. C, Vol. 114, No. 40, 2010, 17023–17029.

We have added this text in the third paragraph of the first Section in Results and Discussion:

“The observation of H₂O at this temperature marks a clear difference to lower pressure studies at UHV conditions, such as the adsorption of water, where desorption is commonly observed at 170K, but here the water is produced in the reaction under steady-state conditions and although the residence time is small, it can still build up high coverage at a high production rate. There are also studies with coverage of water showed significant desorption as high as 400K⁴².”

/3/ The light grey original data points in all XP spectra are hardly visible. In this form the quality of the peak fits cannot be assessed.

Author response:

We have updated all XPS figures and now use red crosses to make the raw data more visible.

/4/ It is not clear how the small fcc fraction of the Co lattice was evaluated from the diffraction data. Please explain.

Author response:

In accordance with the request from the reviewer, we have added a clarification of our argument for the fcc part being small to the SI Section S5. This text now reads:

“The intensity associated with fcc at its maximum is 5000 intensity units. The shape of Bragg reflection suggest that they originate from small fcc domains as their Bragg spot is not very pronounced and they are rather smeared out. Their width is about 20x the width of the CTR of the hcp Co surface. Since we need to account for its width along its orthogonal, we assume it to be similar and thus add another factor of 20x leading to a total intensity of 2×10^6 intensity units. Our observed Bragg peaks of the hcp fraction in the fit lie in the regime of 10^8 to 10^9 which allows us to place an upper border of 1% fcc fraction and thus we deem its contribution small.”

/5/ The O 1s and C 1s signals of gas phase CO are outside the indicated energy ranges. Just for completeness, have the gas phase signals been measured?

Yes, we have measured the gas phase spectra however the information contained is limited and thus we have omitted them from this report. No gas phase products are detected.

We have added a new Section S6.f in the SI to illustrate the gas-phase and adsorbate signal which we commonly detect.

The text reads as follows:

Supplementary Figure 6.1 Gas phase and adsorbate spectrum of the C 1s and O 1s core-level for CO.

“The gas-phase and adsorbate XPS spectrum of carbon monoxide is depicted here under reaction conditions at the indicated pressure and temperature. This is representative of all gas-phase spectra in that we only observe a single peak without an indication of the reaction products. We attribute this to the fact that our reactor has a very short residence time of the gas (~1ms), due to the front cone design, while the reaction is expected to turn over at a rate of 100s per turnover. Thus, the products would be on a scale of 1 per hundred thousand which is lower than typical detection limits for XPS (parts per thousand levels).”

/6/ The introduction, namely lines 49/50 and 59-62, creates the impression that there has been no prior knowledge at all about the chemical state of the Co surface at a syngas pressure of 1 bar. It is clear that STM, one of the previously used methods under reaction conditions, has no direct chemical sensitivity. However, one can often conclude about the surface chemistry from the observed structures. For example, in an operando STM study on the CO oxidation on a Pt(110) surface (Hendriksen et al., Top. Catal. 36, 43, 2005) it was shown that the surface became rough during the reaction which could

be interpreted as the formation of a surface oxide. On the Co(0001) surface, Co carbide would probably have been identified by its characteristic structure, so that its absence indicated that the surface was metallic (refs. 3, 4, 25, and Banerjee et al., J. Phys. Chem. Lett. 7, 1996, 2016). The achievements of the present study are not weakened by these facts.

Author response:

It is a good point that certain chemical states may be inferred from STM and we adjust our text to reflect this possibility. However, as the referee points out the chemical speciation is the strength of XPS.

Text modification in the fourth paragraph of the Introduction:

“Scanning tunneling microscopy (STM) have probed Co single crystal surfaces under FT at atmospheric conditions where the morphology of steps and terraces could be followed but without ~~direct chemical~~ sensitivity towards the reaction intermediates and adsorbates^{3,4}. However, the observed smoothness of the surface in the STM studies indirectly infers that no large rearrangement of substrate atoms has occurred related to oxide or carbide formation “.

/7/ The introduction, lines 77-79, also suggests that infrared spectroscopy under reaction conditions, ~1 bar syngas, was unable to detect anything else than adsorbed CO (based on ref. 24). However, there is a DRIFT study on an oxide-supported Co catalyst at 1 bar, Paredes-Nunez et al., Catal. Today, 242, 178, 2015, presenting spectra that show hydrocarbons and other organic compounds in addition to the CO. These species were probably not adsorbed on the Co but on the oxide support. The fact that they were not seen in work on Co single crystals in ref. 24 (also in Beitel et al., J. Phys. Chem. B, 101, 4035, 1997) could therefore just be explained by the fact that no such species were present. It must not be a limitation of infrared spectroscopy. In fact, the conditions in the Beitel paper, 300 mbar syngas, 490 K, were similar to those in the experiment of Fig. 2b, 200 mbar, 485 K, of the present paper. According to the XPS, mainly adsorbed CO is present but hardly any hydrocarbons.

Author response:

The discussed IR study has clear relevance to the discussion in the introduction and we thank the referee for the suggestion. We have rephrased our text to reflect IR spectroscopy's achievements more clearly.

“There have been efforts to detect adsorbed species with Infrared Spectroscopy but their ~~here~~ the observation is limited to exclusively probing showed adsorbed CO since no other species

in the Co surface region could be detected²⁴ or hydrocarbons that most likely were not on the Co surface²⁵.”

/8/ From S8, it is not clear which physical system was used in the DFT calculations. A layer of molecules without the metal surface?

Author response:

We have clarified in S8 how the DFT system has been calculated. Further we have added a new Supplementary Figure 7 to illustrate the slabs used.

The text in the SI and Figures now reads thus:

*“Density-Functional Theory (DFT) calculations were performed with the GPAW (generalized plane augmented wave)^{S28,29} software using PAW:s. Calculations were done using a slab with four layers of metal atoms where the molecules are adsorbed at the surface. The system is built so the equivalent of ¼ ML coverage is produced as shown in **Error! Reference source not found.** The bottom two metal layers of the slab are frozen with respect to bulk Co while the other degrees of freedom are optimized, so we have a proper relaxation of the molecule adsorbed in different sites and all the metallic atoms surrounding it.*

The finite-difference mode with grid spacing of 0.2 Å was used in conjunction with a Monkhorst–Pack k-point sampling density of 2.5 k-points/Å³ in the supercell and the revised Perdew-Burke-Ernzerhof (RPBE) exchange-correlation functional^{S30}. For binding energy (BE) calculations we must make sure the slab’s supercell is big enough (in our experience larger than 8 Å on each dimension) so core-hole interactions with its periodic images are reduced. The k-point sampling density was also increased to at least 3.5 k-points/Å³.

The absolute XPS BEs are calculated using generated core-ionized PAW:s (explicit core-hole) representing the final or excited state and the BE is obtained from the equation

$$BE = E_{final} - E_{gs}$$

where E_{final} is the total electronic energy of the final or excited state, and E_{gs} is the total electronic energy of the initial or ground state. The BEs are then shifted according to the binding energy position of CO_{top} .”

Supplementary Figure 7. Example of periodic slabs for DFT calculations for CO adsorbed on-top of a Co(0001) surface. On the left we have a more compact supercell that can be used for optimization purpose. On the right the supercell is doubled so we avoid core-hole interactions with its periodic images on BE calculations.

/9/ Minor corrections: Lines 54 - 56: In ref. 12 Co₂C was investigated, not CoC₂.

Author response:

Was corrected.

Lines 65 - 67: "Afterwards such C can either attach to other carbon atoms and thus grow the hydrocarbon chain or ... "; a coupling reaction between C atoms would give chains only consisting of C atoms, not a hydrocarbon chain.

Author response:

We agree our sentences was confusing.

It now reads:

"Afterwards such C can both attach to other carbon atoms as well as adsorbed hydrogen and thus grow the hydrocarbon chain. The final step is the termination of the growth through the attachment to hydrogen atoms that results in enough weakening of the bond between the carbons and the surface, ultimately leading to desorption."

Lines 96 - 98: This sentence seems incomplete.

Author response:

Was corrected.

It now reads:

"Near-ambient XPS (NAPXPS) studies of Co foil have been restricted to 0.1 mbar²³ — far from the conditions of atmospheric pressure where the FT reaction occurs — have detected significant oxidation of the Co foil at low temperatures, while atmospheric pressure single

crystal studies showed the production of methane and other short chain alkanes and alkenes^{3,4,30}.”

Lines 106 - 109: The stepped crystal investigated in ref. 3 was Co(1.0.-1.15), not Co(1.0.-1.5).

Author response:

Was corrected.

Lines 247/248: Fully hydrogen saturated species would be methane, ethane, etc.; the adsorbed hydrocarbon fragments cannot be fully hydrogen-saturated.

Author response:

Was corrected. The text now reads:

“Next component, located at 284.1 eV (light red) we assign to ~~hydrogen-saturated monomers of longer chain~~ hydrocarbon ~~fragments~~, such as -CH₂- and -CH₃ groups on the surface based on DFT calculations and previous post-analysis experiments².”

Further more we have added this sentence in the main text:

“All these species are exemplarily depicted in the Supplementary Information in section S12.”

The Section S12 is shown here:

S12.Schematic Depiction of Possible Adsorbate Configuration

In this section we include Supplementary Figure 2, a graphical schematic depiction of the probable adsorbates in the colours as they are presented to the peaks in the main text for easy recognition for the C 1s core level. In Supplementary Figure 3 we present the same but for the O 1s core-level.

Supplementary Figure 2. Schematic depiction of probable adsorbates for the C 1s core-level.

Supplementary Figure 3. Schematic depiction of probable adsorbates for the O 1s core-level.

Reviewer 3:

In the submission the authors report an operando X-ray photoelectron spectroscopy study of Co(0001) and Co(101 $\bar{4}$) and operando high energy surface X-ray diffraction of Co(0001) during the Fischer-Tropsch reaction at 0.15 bar - 1 bar and 406 K - 548 K in a H₂/CO gas mixture. Some interesting results were observed, such as the atomically smooth metallic Co surfaces under all conditions and the coverage of chemisorbed species ranges from 0.4 – 1.7 monolayers depending on pressure and temperature, a rate-limiting direct CO dissociation pathway with an exclusive participation of hydrocarbon species in the chain growth. These results deepen the fundamental understanding of Co-catalyzed FT synthesis of broad interest and great importance.

The manuscript can be accepted after the authors clarify the following issues:

1. Fundamental studies of Co-catalyzed FT reaction were carried out rather extensively under UHV and NAP conditions. Although the submission report the results under an elevated pressure, most results are similar to those reported previously (such as Ref. 40), and a main contribution of the present work is to demonstrate that the previous results also hold under reaction conditions approaching the working condition more closely. The authors are strongly suggested to discuss their results in the context of previous work appropriately by clarifying this fact.

Author response:

We agree that it is important to discuss the current work in the context of the previous NAP-XPS studies. However, our conclusion is not quite the same as the reviewer suggests. For example, several of the suggestions from the UHV work such as the presence of the ethylidyne intermediates could not be detected in our work. This was pointed this out in the previous text:

“The component observed at 283.5 eV (yellow) we assign to chemisorbed CH₂ species based on DFT calculations (S8). This binding energy has previously been reported as related to the CH₃ group in ethylidyne²⁹ but spectra of the chemisorbed ethylidyne molecule also include a peak corresponding C to the group bonded to the surface at 282.9 eV. Since we observe the 283.5 eV feature at several conditions without any low binding energy component the 283.5 eV peak cannot be associated with ethylidyne. With a similar argument, the 283.5 eV feature cannot be one of the carbons in adsorbed ethylene, where the adsorption site generates two inequivalent C atoms, since then it should be accompanied by a second carbon peak at 283.9 eV²⁹.”

We have added two sentences about the connection to the previous NAP study. Due the space constraints for Nature Communication we cannot go into too many details:

“Several of the current observations in terms of adsorbed CO and presence of different hydrocarbon species on the surface have also been seen in the previous NAP study conducted

at 100-1000 times lower pressures⁴⁶. However, the higher pressure here also resulted in major differences as evidenced by the fact that CO adsorbs in the top site only, significantly smaller C/CH coverage, C_xH_y species observed already at 406K, no oxidation of the surface and no buildup of C at higher temperatures.”

2. It is not appropriate to use the term "operando" because no reaction data together with the in situ characterizations is provided. Were there any gaseous products observed by XPS?

Author response:

See response to reviewers 1 and 2 for the gas products (particularly Reviewer 2 point /5/).

We have rephrased our findings to reflect the in-situ nature of our observation. All claims to the *operando* nature of our findings have been transcribed to *in-situ* including in the title. The title now reads:

***In-situ* Probing of the Fischer-Tropsch Reaction on Co Single Crystal Surfaces up to 1 bar**

We have rephrased the second to last sentence of the Introduction to read:

*“Furthermore, we show the facile appearance and disappearance of C_xH_y adsorbates as seen in section II.d as indicators of a state **where the reaction should occur.**”*

Also, the third to last sentence in the paragraph describing Fig. 3 has been modified to read:

*“Continuing in this configuration, the 284.5eV hydrocarbon peak intensity reduces, indicating facile reaction and departure from the surface aided by the presence of H₂, which supports our **claim that a reaction is ongoing.**”*

3. The arguments of direct CO dissociation and termination step as the rate-limiting steps need to be re-considered. Under the adopted reaction conditions, the reaction seems to be in an equilibrium state which can not be used to justify the rate-limiting step. In the Ref. 2 and 40, the removal of oxygen adatoms resulted from CO dissociation is identified as the rate-limiting step by studies from temperatures starting at RT.

Author response:

We believe that some clarification can be made regarding this point. The state of the reaction is not an equilibrium where the forward reaction and back reaction occur at the same rate. Instead, we are under (or at least very close to) a steady-state condition meaning that the forward reaction is proceeding with constant rate over prolonged period of time. In this situation, certain stable species before the rate controlling steps along the reaction coordinate amass. These species can, in turn, be detected by XPS and thus the XP spectra do indeed yield information about rate limitation.

In the current study we never observed any significant amount of detectable adsorbed oxygen. The study in ref. 2 is not under in-situ conditions and the oxygen could have appeared due to CO dissociation as the hydrogen gas was pumped out limiting the hydrogenation ability. Ref. 40 was under 100-1000 times lower hydrogen pressure. We believe the difference in appearance of adsorbed oxygen is simply the higher hydrogenation ability as FT reactions occurs at high pressures.

We have the added the following sentence in manuscript.

“It has been proposed that also the removal of oxygen atoms is a rate limiting step (ref 2 and 40) but in the current study no adsorbed oxygen was detected. We associate this lack of oxygen compared to the previous studies as most likely due to higher hydrogenation activity in in-situ studies at high pressures.”

Further we have added a sentence in the introduction to clarify the reactions state with regard to equilibria and steady state conditions:

“The virtual pressure cell is established by introducing a high velocity gas jet onto the catalyst and building up a dynamic pressure, such that the gas in contact with the catalyst typically interacts roughly for times on the millisecond scale. This in turn brings the FT reaction over Co into a early steady state, far away from the chemical equilibrium, with low concentrations of products in the effluent gas stream.”

4.What is the definition of monolayer? Please clarify the speciation of adsorbates with the total coverages above 1 ML.

Author response:

We define 1ML to mean that there would be one adatom on every top position on the Co surface. We are showing this now as a graph in the SI S13 Supplementary Figure 12.

We have added the following text in the manuscript.

“The observation of coverages above 1ML signifies that at these conditions we expect an amount of C on the surface able to cover more than all Co surface sites. The increase of atoms on the surface is due to the appearance of hydrocarbon chains linked to the surface but sticking out into the gas phase.”

Further, we added a depiction thereof in the Supplementary Information:

S13.“Schematic Depiction of a Monolayer

In this work we define a monolayer as atoms occupying a site directly on top of all Co atoms making a full hcp layer. We are depicting the side view of this in Supplementary Figure 4.

Supplementary Figure 4. Schematic depiction of a monolayer as used in this work.”

5. About the assignments of C 1s XPS peaks, the results of well-defined alkyl fragments on Co single crystal surfaces (J. Phys. Chem. C 2019, 123, 7740–7748; J. Phys. Chem. C 2020, 124, 24786–24794; Surface Science and Technology (2023) 1: 5) can be cited.

Author response:

We have resolved the recommendations to these articles [10.1021/acs.jpcc.0c07189](https://doi.org/10.1021/acs.jpcc.0c07189), [10.1021/acs.jpcc.8b06010](https://doi.org/10.1021/acs.jpcc.8b06010), [10.1007/s44251-023-00004-7](https://doi.org/10.1007/s44251-023-00004-7).

The articles are quoted in the main text at the appropriate location.

RESPONSE TO REVIEWERS' COMMENTS

We would like to thank the reviewers for taking the time to read the manuscript and provide a more detailed critical review. The comments have helped us develop a stronger and clearer manuscript.

Reviewer 2:

Report on revised ms "Operando Probing of the Fischer-Tropsch Reaction ..." by Lomker et al.

In essence, I am content with the revision of this ms and with the comments to my report. However, I still doubt the interpretation of the O 1s peak at 533 eV as adsorbed H₂O. The authors argue that the presence of H₂O on the Co surface at the high temperature of 406 K can be understood by the reaction, namely by the relatively high H₂O production vs. desorption rate. In my view, this argument is not valid. The production rate of H₂O is identical to the TOF of the Fischer-Tropsch synthesis, which is a well-known quantity. Values given in the review article by Ribeiro et al., Catal. Rev.- Sci. Eng. 39, 49 (1997) vary between 10⁽⁻³⁾ and 10⁽⁻¹⁾ s⁽⁻¹⁾, depending on the sample and experimental conditions. The desorption rate of H₂O can be estimated from the H₂O adsorption energy on Co(0001) for which Weststrate et al., J. Phys. Chem. C, 127, 2974 (2023), measured a value of 57 kJ mol⁽⁻¹⁾. Application of the Arrhenius equation and using a preexponential factor of 10⁽¹³⁾ s⁽⁻¹⁾ gives a desorption rate at 406 K of approximately 5 x 10⁽⁵⁾. This value is more than 6 orders of magnitude higher than the production rate, showing that, at 406 K, H₂O cannot exist on the metallic Co in measurable quantities. The authors cite J. Phys. Chem. C, 114, 17029, 2010 (fig. 8) as an example where H₂O desorption from a Co sample was reported to occur at temperatures as high as 400 K. However, the authors of this paper showed that in this experiment H₂O desorbed from an oxide surface rather than from metallic Co, and that H₂O was strongly bound on the oxide surface. I recommend that the authors mark the 533 eV peak as unexplained; it does not compromise the message of the paper.

Author response:

The observation that the late desorption of H₂O happened on a cobalt oxide surface has eluded us and we apologize for the confusion. We accept the idea that H₂O existence can very well be debated at these conditions and thus mark the peak in our figures adequately as contested by the label “*”. We offer discussion that it could be a consequence of H₂O being present in the purified gases but agree that this peak only has negligible consequence for our manuscript.

We have reformulated a paragraph in “Chemical and Structural State of Co Single Crystals and Adsorbates at 1bar” to read:

The O 1s intensity again shows only significant contributions of CO_{top} adsorbates at 506 K and additionally intensity in the 533.0eV region at 406 K. The origin of the 533.0 eV peak is still an open question. Since we observe no changes in the C 1s

spectrum which follow the variation of the 533.0eV peak we can exclude an origin in oxygenated carbon-containing species. Chemisorbed water, either from contaminations in the purified gas or as a product of the FTS reaction, are other hypotheses which agree with the binding energy value of the peak. The known rapid desorption kinetics of water from Co surfaces—which has been observed even at temperatures as low as 170K [Weststrate et al.]—decreases the likelihood of these hypotheses.

Minor: In the revised version, the authors provide information about the DFT calculations, but they do not say whether these were done spin-polarized (cobalt is magnetic).

Author response:

In the SI to the paper, we have now clarified that Co is calculated in a spin-polarized formalism and the section Supplementary Materials S8 now has this text appended:

Since Co is magnetic, we have used spin-polarization in all calculations. The resulting magnetic moment observable was $1.8 \frac{\mu_B}{\text{atom}}$ well in line with literature [murthyStructurePropertiesEngineering2003 p.381].

The sentence "Near-ambient XPS (NAPXPS) studies ..." is still grammatically incorrect.

Author response:

We have reformulated this sentence into two independent ones and fixed the grammatical error, we thank the referee for noticing. The new text now reads:

Near-ambient XPS (NAPXPS) studies of Co foil have been restricted to 0.1 mbar²³ — far from the conditions of atmospheric pressure where the FT reaction occurs. These studies have detected significant oxidation of the Co foil at low temperatures, while atmospheric pressure single crystal studies showed the production of methane and other short chain alkanes and alkenes^{3,4,30}.

Reviewer 3:

In the revised submission, the authors replied appropriately to most of my previous comments except that the authors still do not consider the adsorption-desorption equilibrium of products on the observed C-containing species in the in situ XPS spectra at low temperatures. The presence of such processes is demonstrated by the observation of adsorbed H₂O species at low temperatures. The neglect of such processes also leads to a puzzling argument that the rate limiting step changes from the formation of carbon chains at lower temperatures to the dissociation of CO at higher temperatures (Lines 382-383). Since the formation of carbon chains occurs after the CO dissociation, I can not see how the rate limiting step changes from the formation of carbon chains at lower temperatures to the dissociation of CO at higher temperatures without considering the facilitated desorption process at high temperatures. After this issue is clarified, the manuscript can be accepted.

Author response:

We realize that there might have been uncertainty in our formulation, especially in the lines 382-383 of the previous manuscript for which we apologize. Of course, we have taken the adsorption-desorption processes and the resulting equilibria into account. We have rewritten that specific paragraph to read:

We notice that the CO dissociation is more facile on the stepped Co single crystal surface. Due to the fast gas exchange rate in the virtual cell (See Supplementary Material section S6.f) accumulation of products in the gas phase and consequentially their reabsorption to the surface are of little significance. On the whole, the increased abundance of hydrocarbon species at 406K on both surfaces, shows that a reaction is ongoing, yet the partly hydrogenated intermediates are not leaving the surface as rapidly as they are being produced. We thus argue that the final termination step in terms of hydrogenation is rate-limiting in this regime. At a temperature of 506K the coverage of hydrocarbons on the surface is lower which indicates that the desorption of products is not limiting the reaction anymore. Since we observe an almost constant CO coverage in all our temperatures the availability of CO is also not a limit in this reaction. Thus, we infer that at this elevated temperature the CO dissociation limits the rate of the reaction to a larger degree.

RESPONSE TO REVIEWERS' COMMENTS

We would like to thank the reviewers for taking the time to read the manuscript and for reviewer 3 to provide a comment review it has helped us develop a stronger and clearer manuscript.

Reviewer 3:

Principally I support the publication of the manuscript. But I would like to encourage the authors to comment on which final termination steps in terms of hydrogenation would overcome barriers higher or as high as CO dissociation in the manuscript, because this finding will be very interesting, and as far as I know, has not been proposed before.

Author response:

We have added a clarification in parentheses displayed in line 415f

(that is exchanging the carbon bond with a local C-H bond)